# Design of Calcium-Binding Proteins to Sense Calcium

**DOI:** 10.3390/molecules25092148

**Published:** 2020-05-04

**Authors:** Shen Tang, Xiaonan Deng, Jie Jiang, Michael Kirberger, Jenny J. Yang

**Affiliations:** 1Department of Chemistry, Center for Diagnostics and Therapeutics and Advanced Translational Imaging Facility, Georgia State University, Atlanta, GA 30303, USA; adatang23@yahoo.com (S.T.); xdeng4@student.gsu.edu (X.D.); jjiang2@gsu.edu (J.J.); 2School of Science and Technology, Georgia Gwinnett College, Lawrenceville, GA 30043, USA; mkirberger@ggc.edu

**Keywords:** rational design, CaBP, protein, protein engineering, synthesis, fusion

## Abstract

Calcium controls numerous biological processes by interacting with different classes of calcium binding proteins (CaBP’s), with different affinities, metal selectivities, kinetics, and calcium dependent conformational changes. Due to the diverse coordination chemistry of calcium, and complexity associated with protein folding and binding cooperativity, the rational design of CaBP’s was anticipated to present multiple challenges. In this paper we will first discuss applications of statistical analysis of calcium binding sites in proteins and subsequent development of algorithms to predict and identify calcium binding proteins. Next, we report efforts to identify key determinants for calcium binding affinity, cooperativity and calcium dependent conformational changes using grafting and protein design. Finally, we report recent advances in designing protein calcium sensors to capture calcium dynamics in various cellular environments.

## 1. Introduction

Ca^2+^ is the most ubiquitous signaling molecule in the human body, regulating numerous biological functions including heartbeat, muscle contraction, neural function, cell development, and cell proliferation. Calcium dynamics result from fluxes in Ca^2+^ concentrations that vary in amplitude and duration, between intracellular compartments and between the intracellular and extracellular space and intracellular organelles [1,2]. Temporal and spatial changes of Ca^2+^ concentrations in different cellular compartments affect the regulation of cellular signaling. Through signal stimulation or alteration in the membrane potential, the cytosolic Ca^2+^ concentration increases from 10^−7^ to 10^−5^ M, and vice-versa. These temporal and spatial changes of calcium concentrations and dynamics are orchestrated by various calcium-binding proteins (Figure 1).

Calcium modulating/triggering proteins such as calmodulin (CaM) are capable of transducing the intracellular Ca^2+^ signal changes into a number of diverse cellular events, including apoptosis, cell differentiation and cell proliferation. Through its cooperative binding to Ca^2+^ via multiple coupled calcium binding sites and calcium-dependent conformational changes, calcium trigger proteins can respond to narrow changes of calcium concentration. Of the myriad Ca^2+^-binding proteins that modulate cell signaling, CaM (Figure 1B) has the most versatile ability to respond to different cellular calcium concentrations through its two pairs of coordinated EF-Hand Ca^2+^-binding sites. Additionally, it activates or inhibits more than 300 functional enzymes, cellular receptors, and ion channels including ryanodine receptor and IP3 receptor at the ER, and gap junction proteins at the plasma membrane, through different binding modes [3,4]. Calcium buffering proteins such as calbindinD_9K_ and parvalbumin act as buffer proteins (Figure 1C) to tightly control intracellular calcium concentrations gradients and kinetics. ER calcium buffer proteins such as calsequestrin with multiple calcium binding sites also controls ER-mediated calcium dynamics. Extracellular calcium concentration is operating at much high calcium concentration (1.1–2.2 mM). G-protein coupled receptors such as calcium sensing receptor and metabotropic glutamate receptors as well as proteins are able to bind to calcium and conduct/trigger subsequent signaling cascade [5,6,7,8]. Calcium binding also plays important structural roles for protein stabilization (Figure 1D), virus assembly and folding (Figure 1E), and to inhibit degradation [9]. However, the mechanisms through which different classes of proteins are able to bind calcium with different affinities (Figure 1A), and the resulting calcium dependent conformational changes, remain poorly understood. Efforts to design CaBP’s were hindered by the diversified coordination properties and binding geometries of CaBP’s, coupled with conformational changes of the proteins due to electrostatic interactions. In addition, there is an emerging need to develop calcium sensors capable of monitoring calcium dynamics and activity in different cellular environments to better understand the role of calcium signaling at cellular and molecular levels.

In this paper, we will first review coordination properties of calcium binding proteins as evaluated by statistical analysis, and the application of protein design to understand the key determinants for calcium binding and its role in control of biological functions. We will then discuss applications of protein design and the engineering of calcium binding proteins for the development of protein calcium sensors capable of monitoring calcium dynamics. For the applications to development of protein MRI (Magnetic Resonance Imaging) contrast agents for molecular imaging, please see other reported reviews [10,11,12,13,14,15].

## 2. Classification and Prediction of Calcium Binding Sites in Proteins

To capture common features of calcium binding proteins and with a goal of visualizing the role of Ca^2+^ in biological systems (Calciomics), we performed a statistical analysis of various types of known Ca^2+^ binding proteins including EF-hand and non EF-hand proteins, and compared these results with properties of small molecules [16,17,18]. Results of these studies demonstrated that the structures and coordination geometries surrounding Ca^2+^ binding sites are more diverse than previously believed, and that Ca^2+^ binding sites in proteins are better reclassified as holospheric and hemispheric, instead of the classical polygon coordination reported in small molecule chelators. Ca^2+^ binding preferences of different amino acids were defined based on their observed frequency within the Ca^2+^ binding sites of proteins. Subsequently, several graph theory-based computational algorithms were developed that utilized our statistical data to predict Ca^2+^-binding sites with high accuracy and sensitivity in proteins using oxygen and carbon clusters [19,20,21,22]. Similarly, our lab has developed a search engine for predicting various classes of calcium binding proteins based on sequential pattern recognition [18,23]. These developed computer methods have been applied to predict calcium binding sites in bacteria and virus systems [9,24], as well as membrane receptors and transporters [8,25,26].

## 3. Development of Grafting and Design Approaches to Discover Key Factors that Contribute to Ca^2+^ Binding Affinities and Selectivities, and Ca^2+^-Induced Conformational Change

To overcome the complexities encountered in cooperative, multi-site binding associated with natural calcium binding proteins, two novel approaches (designing and grafting) have been developed to create single Ca^2+^-binding site sensor proteins. These structure present opportunities to dissect the key structural factors responsible for Ca^2+^-binding affinity, conformational change, and cooperativity (Figure 2).

A single continuous calcium binding site like the EF-loop of CaM can be engineered into the loop region of a scaffold protein, like domain 1 of cell adhesion molecule CD2, which can tolerate mutations and has Trp residues. A flexible linker such as GGG is used to flank the engineered calcium binding site, which enables the Ca^2+^-binding loop to retain its native conformation for calcium binding. Intrinsic calcium binding affinity of the continuous Ca^2+^-binding site can be determined by bacterial expression and purification of the grafted protein. Tb^3+^, which has coordination properties similar to Ca^2+^, can be used to determine Tb^3+^ binding affinity via Tb-sensitized energy transfer. Calcium binding affinity is then determined by competition with the FRET enhancement. This method is very sensitive as the three positive charges on the Tb^3+^ ion exhibit higher metal binding affinity than Ca^2+^, and the use of Tb^3+^ has additional advantages over Ca^2+^ which often causes interference due to its ubiquitous background presence in solutions. Using this grafting approach we reported the first estimation of the intrinsic Ca^2+^ affinities of the four EF-hand loops of CaM, and a better estimation of the cooperativity of coupled EF-hand proteins [27,28,29,30]. This developed grafting approach has been successfully applied to obtain site-specific calcium binding affinities of our predicted single EF-hand motif in the non-structural protein of rubella virus [24]. Later, we further extended this grafting approach to probe site specific calcium binding affinity of non-EF-hand continuous calcium binding sites in G-protein coupled receptors (GPCR) including calcium sensing receptor [25,26] and mGluR1 [8]. Taking advantage of its relaxation properties, the lanthanide binding capability of EF-hand peptides have also been used to determine NMR (Nuclear Magnetic Resonance) structures [31].

Using identified common calcium coordination properties in proteins based on the statistical analysis and developed prediction algorithms, we successfully designed several de novo calcium-binding sites into the non-calcium binding protein CD2 (Figure 3) [32]. The NMR solution of the structure (1T6W) revealed that Ca.CD2 would bind Ca^2+^ in the designed site, which validated this general strategy for de novo design of Ca^2+^-binding proteins [33,34]. We have shown that Ca^2+^ and Ln^3+^-binding affinities, and protein stability, can be systematically tuned by altering metal coordination shells and thus the protein environment [35,36,37]. Using a single designed calcium binding site in the scaffold protein did not produce large conformational changes, thereby demonstrating that calcium binding affinity can be gradually increased by increasing the number of charged residues in the coordination shell from three to five. Additionally, to understand calcium dependent conformational changes, we also successfully designed a Ca^2+^-dependent CD2.trigger whose conformation could be reversibly switched upon Ca^2+^-binding without using coupled EF-hand motifs [38].

## 4. Development of Genetically Encoded Calcium Indicators (GECIs) Using Protein Engineering

Building on the earlier work using aequorin to sense calcium, fluorescent proteins with intrinsic chromophore (FPs) of different colors have been developed for a variety of purposes, including uses as sensors, indicators of activity, and for enhanced microscopy visualization in cells and in tissues [39,40,41,42]. Genetically encoded calcium indicators (GECIs) have been widely used to elucidate various physiological mechanisms regulated by calcium signaling. Advanced GECIs provide quantitative analysis of Ca^2+^ fluctuations in different subcellular compartments which is essential to defining the mechanisms of Ca^2+^-dependent signaling under physiological and pathological conditions. Here we review the approaches for development of current GECIs and provide potential strategies for future development.

The endogenous Ca^2+^-trigger protein calmodulin (CaM), or troponin C (TnC), is commonly used as the calcium-binding domain to sense calcium changes for the design of most GECIs. Following cooperative binding to calcium, CaM or TnC exhibit additional conformational changes upon binding to its target proteins or peptides such as M13 peptide sequence from myosin light chain kinase (MLCK). This conformational change was first coupled with distance changes of two fluorescent proteins that have overlapping fluorescence excitation and emission wavelengths, producing a Forster resonance energy transfer (FRET) change. The development of calcium sensors was pioneered by Persichini [43] and Roger Tsien [1], whose subsequent work on engineering changes to GFP [44] led to a wide range of fluorescent protein applications. Improvements were then made to reduce interferences with intracellular CaM or CaM targeting proteins by modifying the CaM/peptide binding surface [45], or by modifying TnC to have narrowed specificity for its target protein TnI [46]. More recently, Barykina et al. reported development of a novel green Ca^2+^ indicator, FGCaMP, comprised of fungal CaM fused with EGFP, and a fragment of CaM-dependent kinase [47]. This novel green Ca^2+^ indicator was utilized to visual calcium transients in mammalian cells and observe calcium dynamics in neuronal cells with significantly improve mobility compared with other GECI’s.

To overcome limitations of FRET-based calcium sensors, Nakai et al. pioneered the development of a GCaMPs series of calcium sensors by leveraging Ca/CaM and its large conformational change binding to the targeted peptide M13 peptide using a single fluorescent protein [48]. CaM was inserted into a circular permeable FPs with M13 fused to the terminal. Since 2001, significant progress has been made via selection to improve the dynamic range and the signal to noise ratio (SNR) of GCaMPs, with 5–10× increases in the dynamic range observed approximately every 5 years. Current GCaMPs exhibit a broad range of binding affinities from sub µM to hundreds of µM. GCaMPs have been broadly applied across a wide range of species for both in vitro and in vivo imaging and also with different colors [49,50,51,52,53,54,55,56,57,58,59,60,61,62].

The upper limit rate for endogenous Ca^2+^ signaling is not known, and it remains to be determined whether the current GECIs are fast enough to reliably report cellular Ca^2+^ signaling in excitable cells during electrophysiological stimulation. Small molecule Ca^2+^ indicators such as OGB-1, which are known for their fast kinetics, should satisfy these requirements. OGB-1 had a k_on_ rate of 4.3 × 10^8^ s^−1^ M^−1^, and a k_off_ rate of 10^3^ s^−1^, thus it was estimated that OGB-1 should be able to report Ca^2+^ spikes up to 10 Hz. However, rather than distinguishable peaks, Ca^2+^ signals measured by OGB-1 appeared as a smooth curve at 10 Hz stimulation [63]. When the stimulation stopped, the peak to basal decay phase of the Ca^2+^ signaling was approximately 2 s, whereas the decay phase of OGB-1 was as short as 5 ms, measured by stopped-flow [64].

In the past several years, considerable effort has been devoted to overcoming the limitations of CaM/TnC based sensors, including slow response kinetics and the stepwise calcium response due to multiple calcium binding sites in GCaMPs, especially for quantification of high calcium concentrations in organelles such as the ER, which exhibits rapid complicated calcium dynamics. Since the initial development of GCaMP-1, newer versions have exhibited significant improvements in optical properties. Red fluorescent proteins (mApple and mRuby)-based GCaMPs were reported, and these exhibit rapid kinetics, making them useful for neuronal imaging. f-RGECO1, f-RGECO2, f-RCaMP1 and f-RCaMP2 are four recently reported variants [65]. The Ca^2+^ dissociation (k_off_) rates of these sensors have been increased by as much as 10 times the rates of the original version. To improve the sensor kinetics, one EF-hand in the calmodulin (CaM) sensor (either EF-1 or EF-4) was mutated to prevent Ca^2+^-binding, and the CaM interaction residue in the targeting binding peptide RS20 was mutated to decrease the affinity between Ca^2+^-loaded CaM and RS20 peptide. The overall Ca^2+^ dissociation constants of these mutated sensors were significantly increased, accompanied by dramatic improvements of the k_off_ rates. Both the k_on_ and k_off_ rates of the sensors could be further increased by raising the temperature. At average physiological temperature (37 °C), the k_off_ and k_on_ rates were as much as 4 and 10 times higher than at 20 °C. The fastest k_off_ rate was reported to be 10^9^ ± 1 s^−1^ for f-RGECO1, which was comparable to the k_off_ rate (10^3^ s^−1^) for the small molecule-based Ca^2+^ indicator OGB-1. Another approach involved modification of NTnC GECI, which exhibits limited response to Ca^2+^, by replacing mNeonGreen FP with enhanced yellow fluorescent protein (EYFP). The new GECI, YTnC, exhibited positive and 3-fold improved response to Ca^2+^, with 4-fold faster kinetics, than NTnC [66]. Results of this study demonstrated that YTnC could successfully visualize calcium transients in neurons of mice using single-and two-photon microscopy, and it outperformed GCaMP6s in HeLa and neuronal cells.

Shen et al. also reported a GECI with fast kinetics and high sensitivity to Ca^2+^ in Hela and neural cells, as well as mouse brain in vivo [67]. This indicator, K-GECO1, combined CaM with a circularly permutated RFP eqFP578 scaffold, and relies on the ckkap peptide as the CaM binding partner, resulting in a lower apparent K_d_ and an apparent Hill coefficient approaching 1. Two recent studies further reported development of GECI’s with fluorescence in the near-infrared (NIR) region of the spectrum. NIR-GECO1, reported by Qian et al. combined CaM with the NIR fluorescent biliverdin (BV)-binding fluorescent protein mIFP [68]. NIR-GECO1 was able to image Ca^2+^ transients in both mammalian cells and brain tissue, but it did not demonstrate the ability to image neuronal activity in vivo. To overcome some limitations identified with NIR-GECO1, Subach et al. designed an NIR GECI, GAF-CaMP2, by combining bacterial phytochrome GAF-FP and calmodulin/M13-peptide [69]. GAF-CaMP2 exhibited a positive response to Ca^2+^, as opposed to the negative response reported for NIR-GECO1, with high Ca^2+^ affinity (K_d_ of 466 nM), and the ability to visualize calcium transients in mammalian cell organelles. Both NIR-GECO1 and GAF-CaMP1 extended GECI responses into the NIR region of the spectrum, resulting in potential applications for multi-color imaging. Thus, it is possible that advanced GECI will exhibit fast kinetics similar to those observed with calcium dyes used for in vivo imaging.

Troponin C-based GECIs with reduced Ca^2+^ binding sites have also been designed to reduce the Hill coefficient to 1 so that they exhibit a linear response for quantification [70,71]. Troponin C can be divided into N- and C- domains, and each domain includes two EF-hand Ca^2+^-binding motifs. The EF-hand sites in the N-terminal have weak binding affinities to Ca^2+^ whereas these at C-terminal can bind Ca^2+^ with an overall sub-µM K_d_. Thus, deletion of the *N*-terminal domain of troponin C could generate a novel sensor with strong binding affinity and a low hill coefficient. To further improve the metal selectivity of this novel sensor, the Ca^2+^ binding residues Asn15 and Asp17 were changed to Asp15 and Asn17 via site-directed mutation, which completely eliminated the Mg^2+^-induced fluorescence intensity changes. This class of GECIs, called Twitch, maintained strong binding affinity to Ca^2+^ (K_d_: 150–450 nM) which was suitable for detection of cytosolic Ca^2+^ signaling [72]. To further reduce the number of Ca^2+^ binding sites, single EF-hand based GECI were created by only remaining either the EF-hand-III or EF-hand-IV site of Twitch. These new sensors exhibited FRET responses acceptable for cellular Ca^2+^ imaging, hill coefficients very close to 1, and a wide range of Ca^2+^ binding affinities from 2.6 μM to 257 μM, thereby making them suitable for measuring high Ca^2+^ concentrations found in Ca^2+^ stores such as the endoplasmic reticulum (ER). In vivo imaging to dissect the relationship between segmental Ca^2+^ dynamics and behaviors under physiological and pathological conditions has been challenging due to continually recording scattering and autofluorescence. To increase the sensitivity and kinetics, G-CaMP3 and other G-CaMPs variants have been faithfully reported neuronal activity in various organisms [50,51]. TC-nano with strong binding affinity was reported to visualize neuron activities in zebrafish embryos [73]. Transgenic Thy1-TwitchER mice were generated for in vivo ER Ca^2+^ imaging by using TwitchER sensors. This sensor expressed well in the dorsal root ganglion (DRG) neurons and exhibited large dynamic ranges in vivo. Either thapsigargin- or caffeine-induced ER calcium depletion could result in over 50% of the FRET ratio change, and this response range was comparable to any other high-sensitive ER sensors [70]. CaMPARI, using photoconvertible fluorescent protein as scaffolds, enable integration of neuronal activity under different thresholds [74,75].

## 5. Development of Calcium Sensors Using Rational Design

To overcome limitations associated with slow kinetics and non-linearity to CaM, Yang et al. pioneered another novel way to create calcium sensors by designing a single calcium binding site into a fluorescent protein without using endogenous Ca^2+^ binding proteins [76,77,78,79,80]. They first grafted a single EF-hand calcium binding motif to EGFP to create a calcium sensor Ca-G1-ER. The subsequent addition of calcium resulted in changes to both the absorbance and fluorescence emission spectra [80]. Detailed kinetic studies revealed that an intermediate state limits kinetics. To overcome rate limited steps by the formation of intermediate complex, CatchER, a calcium sensor for detecting high calcium concentrations, was generated by mutating five solvent-accessible residues on the surface of EGFP but in a location opposite to the chromophore, into negatively-charged residues, thereby forming a new Ca^2+^ binding site (Figure 4). Binding of Ca^2+^ to this site then triggered local conformational change to alter the hydrogen bonds surrounding the chromophore, resulting in changes to the fluorescent intensity. CatchER produced two absorption peaks at 395 nm and 488 nm. The Ca^2+^ binding process resulted in increased intensity of the 488 nm peak, with concurrent loss of signal intensity of the 395 nm peak. Additionally, excitation at either 395 nm or 488 nm resulted in higher peak intensity at 510 nm in the Ca^2+^ loaded form compared to the Ca^2+^ free form. Wild type GFP exhibits peaks at both 395 nm and 488 nm, while EGFP produces a single peak at 488 nm. The high-resolution X-ray crystal structure of CatchER suggested that the hydrogen network of the apo-form chromophore was similar to wild type GFP, whereas the Holo form appeared to mimic EGFP. These data suggested that the slight rotation of the Glu222 sidechain that occurred during transition from the Apo- to the Holo- form of CatchER played an important role in altering the chromophore hydrogen bonds.

CatchER has been applied to understanding ER calcium releases in EC-coupling and molecular mechanism related to muscle fatigue due to aging [76]. Interestingly, CatchER also exhibited a fluorescence lifetime change using a time-correlated single photon counting (TCSPC) system at emission wavelength 510 nm following excitation at 372 nm and 440 nm, respectively [79]. There was a FLIM increment (~0.4 ns) with CatchER at 372 nm following saturation with Ca^2+^, which was not observed at 488 nm excitation. Decreasing absorption at 395 nm with a concomitant emission increase at 510 nm resulted in an increase in quantum yield, and an associated increase in FLIM, after binding Ca^2+^. When the pH changed from 5.0 to 9.3, the absorption spectra of CatchER exhibited a ratiometric change similar to what was observed after adding Ca^2+^, where the 488 nm peak increased while the 395 nm peak decreased, and the emission intensity at 510 nm increased when excited at both of these two wavelengths. This spectra change also produced a similar quantum yield change, and the FLIM change further verified this mechanism. There was a dramatic FLIM increment at 372 nm excitation when the pH changed from 5.0 to 9.3, consistent with the results from the Ca^2+^ titration process. The FLIM imaging exhibited advantages in quantitative Ca^2+^ imaging, especially in high scattering samples, and the single fluorescent protein-based CatchER presented a unique FLIM mechanism. This protein design approach in creating calcium sensor has advantages in tuning calcium affinities and rapid kinetics by modifying the calcium coordination and minimizing perturbations in the alteration of endogenous calcium signaling. CatchER is expected to have broad biological applications (Figure 4).

## 6. Perspectives, Future Research, and Conclusions

Currently there are many exciting advances underway that involve using protein design and engineering of calcium binding proteins. One exciting development involves efforts to map proteins and various biological processes regulated by calcium using identified key determinants in calcium binding and calcium dependent conformations (Calciomics [81]). Taking advantage of the significant amount of structure, proteomic and genomic information and data, major biological pathways and molecular mechanisms will likely be visualized. In addition, continuing research directed at elucidating the structures of calcium binding proteins and understanding their functions will also provide new insights into various pathological states. The ability to modify proteins allows us to produce modifications for the purpose of research into the mechanisms of disease, toxicity, cellular function, and genetic disorders. Building on these development, molecular basis of diseases associated to calcium binding and handling of calcium binding proteins such as calcium sensing receptor, calmodulin and gap junctions are currently important topics of research [7,25,26,78,82,83,84,85,86,87,88,89,90,91,92,93,94]. We have also demonstrated that RyR can be regulated by manipulating calcium binding affinity of CaM [95,96]. Further, great effort will likely to devote to design multi-color calcium sensors with event fast kinetic responses and calcium affinity tailored to various cellular organelles to provide a comprehensive map of calcium dynamics. It is also important to develop calcium sensors with various targeting capability to tissue, organs and animals for in vivo monitoring calcium dynamics. While limited by protein immunogenicity due to florescent proteins are originated from non-human resources, in vivo imaging in various animal models, plants and ex vivo application in organs are likely to help to unveil the “calcium code” in its capability to regulate numerous biological proteins and facilitate to drug discovery. The sheer magnitude and diversity of proteins that do or can exist represents an important resource that can be explored through biotechnology for the development of modified biomolecules with new functions, including new materials for diagnostic and therapeutic applications [10,11,12,13,14,15]. Furthermore, using these established design strategies, we have created novel protein based MRI contrast agents for molecular imaging of an array of disease biomarkers with strong translational potentials. Many of these have been reported in related reviews [10,11,12,13,14,15], including the use of calcium binding proteins such as calmodulin to design enzymes [97,98]. Metalloproteins and binding motifs for different metals, like Ca^2+^ and Zn^2+^, may be modified to selectively bind other metals, potentially leading to new advances in chelation therapy, imaging diagnostics, and therapeutic agents designed to combat invasive cancers. The inclusion of targeting moieties in these chimeric protein structures can facilitate direction to specific antigens and biomarkers, so that therapeutic effects can be focused with greater precision. These approaches will likely yield new and improved treatments for different types of cancers, offering new hope to patients from treatment options that are less toxic and destructive than radiation therapy or traditional chemotherapies. The examples presented in this review represent not only our current state of progress, but insight into the potential opportunities that remain to be discovered.

## Figures and Tables

**Figure 1 molecules-25-02148-f001:**
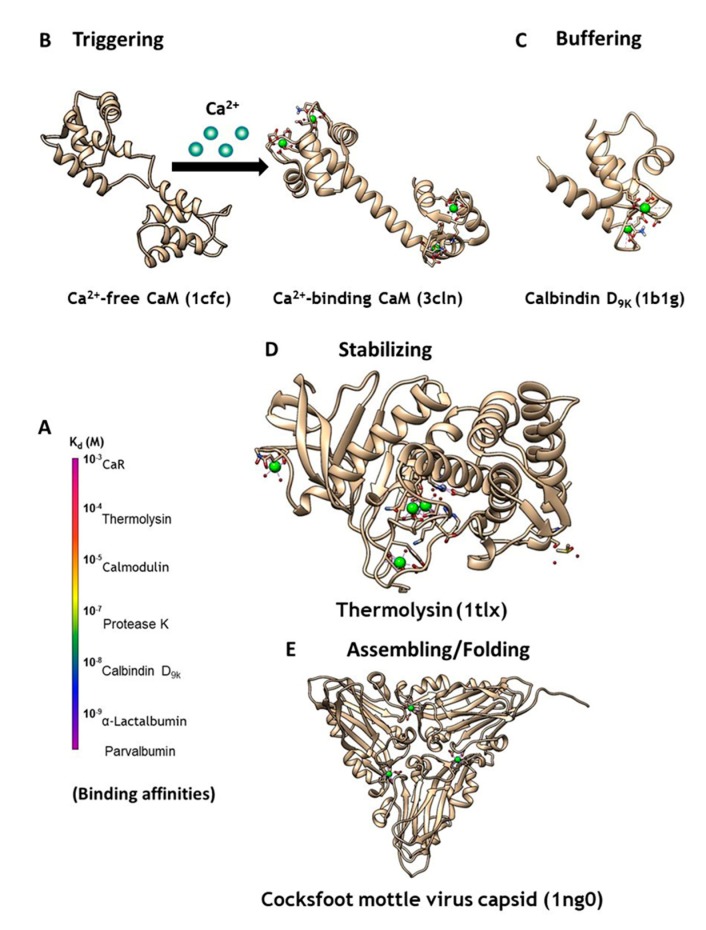
(**A**) CaBPs exhibit a broad range of binding affinities, which can vary by function. (**B**) Intracellular signaling protein calmodulin (CaM) binds up to four calcium ions as a secondary messenger. (**C**) Calbindin D_9K_ acts as an intracellular calcium buffering protein. (**D**) Thermolysin is a zinc-activated metalloproteinase that is stabilized through binding of four calcium ions. (**E**) Calcium may also be required as a cofactor for assembling and folding of virus capsid, as seen in the Cocksfoot mottle virus.

**Figure 2 molecules-25-02148-f002:**
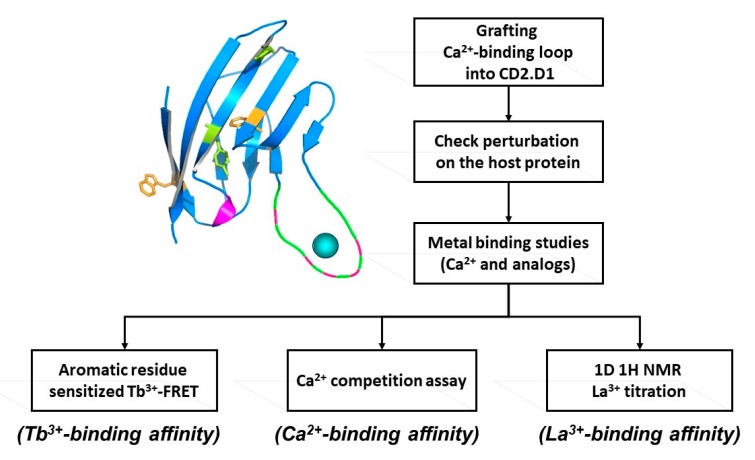
Determining intrinsic metal binding properties stepwise, through grafting of EF-loop (top left) into a scaffold protein (top right).

**Figure 3 molecules-25-02148-f003:**
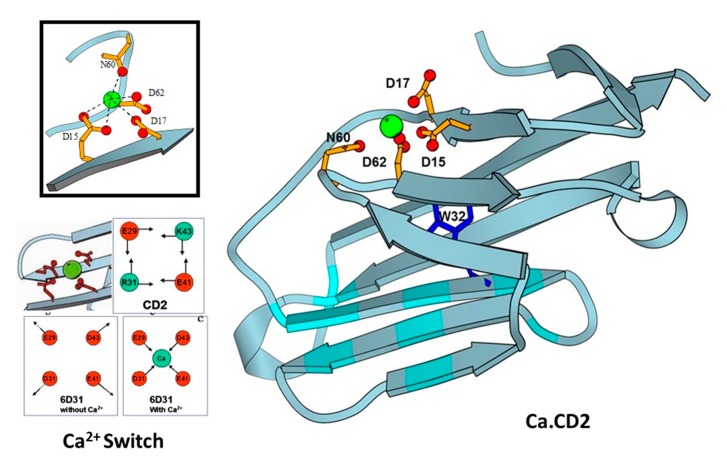
Design of CaBPs using cell adhesion molecule CD2 domain 1 with different arrangement of charged ligand residues. Determined NMR structure for Ca.CD2 (1T6W). Affinties were charge-dependent, with the highest affinity corresponding to the highest negative charge (−5 > −4 > −3 > −2).

**Figure 4 molecules-25-02148-f004:**
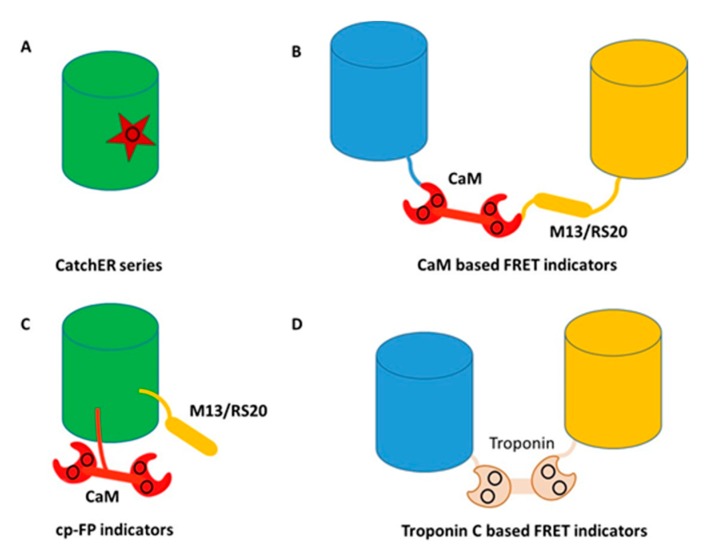
Classes of genetically encoded calcium indicators (GECIs) (**A**) rational design of a single calcium binding site, CatchER series. Star represents the binding site consisting of negatively charged residues; (**B**) CaM based FRET indicators. Blue and yellow cylinders represent blue and yellow fluorescent proteins, respectively. The black circle shows the calcium binding sites; (**C**) CaM based single wavelength indicators. The black circle shows the calcium binding sites; (**D**) Troponin C based FRET indicators. Blue and yellow cylinders represent blue and yellow fluorescent protein, respectively. The black circle shows the calcium binding sites.

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
