# Peer review of "Design of Calcium-Binding Proteins to Sense Calcium"

_molecules, 2020, doi:10.3390/molecules25092148_

Round 1

Reviewer 1 Report

In this manuscript Shen Tang et al. reviewed current progress in the field of the rational design of calcium binding proteins and genetically encoded calcium indicators. Authors reported different types of calcium binding proteins with a broad range of calcium binding affinities and functions. Shen Tang et al. described computational approaches used for prediction of calcium binding sites in proteins. Authors next overviewed grafting and design strategies for de novo engineering and modification of single calcium binding site. Finally, Shen Tang et al. gave description of the current progress in the development of the fluorescent genetically encoded calcium indicators. Overall, this review provides insight into current state of progress and future potential opportunities in the calcium binding proteins design and related diseases research and treatment. 

Minor points:

Point 1. Lines 31-32. I would recommend to add “and vice versa” at the end of sentence. 

Point 2. Line 69 and 114. MRI and NMR abbreviations needs to be defined.

Point 3. Lines 142-161. In this paragraph utilization of calmodulin from fungus in FGCaMP GECI should be mentioned (ref. Barykina, N.V. et al. Plos One 2017).

Point 4. Line 176. K-GECO indicator based on FusionRed protein (from Campbell R group) should be mentioned (ref. Shen, Y. et al. BMC biology 2018). Also, near-infrared calcium indicators based on bacterial phytochromes should be described (such as NIR-GECO1 ref. Qian, Y. et al. Nature methods 2019 and GAF-CaMP2 ref. Subach, O.M. et al. Int J Mol Sci. 2019).

Point 5. Line 190-191. YTnC GECI based on the YFP protein with inserted TnC should be mentioned (ref. Barykina, N.V. et al. Scientific Reports 2018).

Overall, if points raised above will be adequately addressed, the publication in Molecules MDPI journal is appropriate.

Author Response

Comment 1: Point 1. Lines 31-32. I would recommend to add “and vice versa” at the end of sentence. 

Our Response: We have added the recommended text.

Comment 2: Point 2. Line 69 and 114. MRI and NMR abbreviations needs to be defined.

Our Response: We have added the recommended text.

Comment 3: Point 3. Lines 142-161. In this paragraph utilization of calmodulin from fungus in FGCaMP GECI should be mentioned (ref. Barykina, N.V. et al. Plos One 2017).

Our Response: We thank the reviewer for directing our attention to this reference. As suggested, we have included the following revision to the manuscript beginning on Line 157:

“More recently, Barykina et al reported development of a novel green Ca2+ indicator, FGCaMP, comprised of fungal CaM fused with EGFP, and a fragment of CaM-dependent kinase [48]. This novel green Ca2+ indicator was utilized to visual calcium transients in mammalian cells and observe calcium dynamics in neuronal cells with significantly improve mobility compared with other GECI’s”.

Comment 4: Point 4. Line 176. K-GECO indicator based on FusionRed protein (from Campbell R group) should be mentioned (ref. Shen, Y. et al. BMC biology 2018). Also, near-infrared calcium indicators based on bacterial phytochromes should be described (such as NIR-GECO1 ref. Qian, Y. et al. Nature methods 2019 and GAF-CaMP2 ref. Subach, O.M. et al. Int J Mol Sci. 2019).

Our Response: We again thank the reviewer for directing our attention to other relevant references. We have included the following text beginning on Line 203:

“Shen et al reported a GECI with fast kinetics and high sensitivity to Ca2+ in Hela and neural cells, as well as mouse brain in vivo [67]. This indicator, K-GECO1, combined CaM with a circularly permutated RFP eqFP578 scaffold, and relies on the ckkap peptide as the CaM binding partner, resulting in a lower apparent Kd and an apparent Hill coefficient approaching 1. Two recent studies further reported development of GECI’s with fluorescence in the near-infrared (NIR) region of the spectrum. NIR-GECO1, reported by Qian et al, combined CaM with the NIR fluorescent biliverdin (BV)-binding fluorescent protein mIFP [68]. NIR-GECO1 was able to image Ca2+ transients in both mammalian cells and brain tissue, but it did not demonstrate the ability to image neuronal activity in vivo. To improve on some limitations identified with NIR-GECO1, Subach et al designed an NIR GECI, GAF-CaMP2, by combining bacterial phytochrome GAF-FP and calmodulin/M13-peptide [69]. GAF-CaMP2 exhibited a positive response to Ca2+, as opposed to the negative response reported for NIR-GECO1, with high Ca2+ affinity (Kd of 466 nM), and the ability to visualize calcium transients in mammalian cell organelles. Both NIR-GECO1 and GAF-CaMP1 extended GECI responses into the NIR region of the spectrum, resulting in potential applications for multi-color imaging”.

Comment 5: Point 5. Line 190-191. YTnC GECI based on the YFP protein with inserted TnC should be mentioned (ref. Barykina, N.V. et al. Scientific Reports 2018).

Our Response: To address the reviewers suggestion, we have included the following text beginning on Line 197:

“Another approach involved modification of NTnC GECI, which exhibits limited response to Ca2+, by replacing mNeonGreen FP with enhanced yellow fluorescent protein (EYFP). The new GECI, YTnC, exhibited positive and 3-fold improved response to Ca2+, with 4-fold faster kinetics, than NTnC [67]. Results of this study demonstrated that YTnC could successfully visualize calcium transients in neurons of mice using single-and two-photon microscopy, and it outperformed GCaMP6s in HeLa and neuronal cells”.

Reviewer 2 Report

Manuscript: Design of Calcium-binding Proteins to Sense Calcium

Calcium controls numerous biological processes by interacting with different classes of

calcium binding proteins (CaBP’s), with different affinities, metal selectivities, kinetics, and calcium dependent conformational changes. Due to the diverse coordination chemistry of calcium, and complexity associated with protein folding and binding cooperativity, the rational design of CaBP’s was anticipated to present multiple challenges. In this paper advances in

statistical analysis of calcium binding sites in proteins and subsequent development of algorithms to predict and identify calcium binding proteins was performed along with identification of key determinants for calcium binding affinity, cooperativity and calcium dependent conformational changes using grafting and protein design. Manuscript needs to be revised based on outlined below comments:

General Remarks:

In the abstract, “In this paper we will first review our advances in statistical analysis of calcium binding …” what does our mean? authors just reviewed their own work?

Figure 1 and 2 are hard to read, please improve the quality

Please make sure to include/cover recent 2019-2020 papers

Is the sensor investigated in human trials? This may need to be discussed

I think additional section about future research/perspectives could be useful

Author Response

Comment 1: In the abstract, “In this paper we will first review our advances in statistical analysis of calcium binding …” what does our mean? authors just reviewed their own work?

Our Response: We thank the reviewer for bringing to our attention the inaccuracies in the abstract and have modified the text to provide improved clarity.

Comment 2: Figure 1 and 2 are hard to read, please improve the quality

Our Response: Both Figure 1 and Figure 2 have been replaced with higher resolution images and have been rearranged to improve visual quality.

Comment 3: Please make sure to include/cover recent 2019-2020 papers

Our Response: As per both reviewer 1 and reviewer 2, we have expanded our manuscript to include more recent work that was brought to our attention including Shen, Y. et al. BMC biology 2018; Barykina, N.V. et al. Plos One 2017; Barykina, N.V. et al. Scientific Reports 2018; Qian, Y. et al. Nature methods 2019 and Subach, O.M. et al. Int J Mol Sci. 2019.

Comment 4: Is the sensor investigated in human trials? This may need to be discussed

Our Response: To our best knowledge, there is no any sensor in human trial for in vivo application due to limitations such as immunogenicity originated due to none of the florescent proteins are human proteins.  It is, however, likely be useful for ex vivo diagnostics and facilitate to drug discovery.

Comment 5: I think additional section about future research/perspectives could be useful.

Our Response: We have strengthen the writing for the future research /perspectives in the last paragraph.

Round 2

Reviewer 2 Report

Accept